# Effects of Cane Emergence Time, Bending, and Defoliation on Flowering and Yield in Primocane-Fruiting Blackberry

**Fumiomi Takeda [1,*], Ann Rose [1] and Kathleen Demchak [2]**

[1] Appalachian Fruit Research Station, 2217 Wiltshire Road, Kearneysville, WV 25430, USA; annrose@peoplepc.com

[2] Department of Plant Science, The Pennsylvania State University, University Park, PA 16802, USA; efz@psu.edu

* Correspondence: fumi.takeda@usda.gov; Tel.: +1-304-725-3451

**Abstract:** Primocane-fruiting (PF) blackberries are adaptable to different production systems. To increase yields in PF blackberries, their primocanes are typically tipped or topped in summer to encourage branch formation from axillary buds below the cut. In this study, we determined in PF 'Prime-Ark® Traveler' whether early emerging primocanes were more productive than those that emerged later in the season, and the effect of primocane bending and defoliation on flowering. The primocanes that emerged in April produced 64% more flower shoots than those that emerged after May. Also, these findings indicate the alternative primocane management practices of selecting the early emerging primocanes and bending to orient primocanes horizontally, and leaf removal to increase budbreak and flower shoot emergence. The present work contributes toward a better understanding of primocane emergence time and orientation–flowering relations, and how these factors mediate crop performance of PF blackberry.

**Keywords:** *Rubus*; cultural practice; leaf removal; flowers; fruit; floricane; trellis; cane training; pruning; management strategy

## 1. Introduction

In North America, thornless blackberry became available with the introduction of 'Smoothstem' and 'Thornfree' by the USDA [1], and now, most blackberry cultivars grown for fresh pack and consumption are thornless. More recent improvements in cultivar development are the primocane-fruiting (PF) trait [2] and innovative production systems for floricane-fruiting (FF) blackberries [3]. These advances have expanded the commercial production in Central America and other areas with mild winter conditions, and in high-latitude regions with low winter temperatures. These changes in FF and PF blackberry production (e.g., new cultivars and production practices) have enabled growers and packers to ship high-quality fresh blackberries to distant markets almost year-round in North America [2,4].

The above-ground portion of blackberry plants consists of canes which emerge from adventitious buds on the perennial root system and latent buds on the crown. Blackberry plants that produce canes that are stout and upright are called the erect type, and those that produce non-erect canes and tend to grow on the ground, if not trellised, are called the trailing type. The canes that develop from the crown and the roots are biennial; thus, mature blackberry plants have two types of canes. The primocanes are first-year canes and are usually vegetative in FF blackberry. Flower bud initiation in FF blackberry occurs from late summer and®may continue into spring of the following year,

depending on temperatures during winter [5,6], and bloom occurs in spring. Over-wintered canes that flower in spring and produce fruit in year two of the cane's life cycle are called floricanes.

In the last 15 years, a series of primocane-fruiting blackberry cultivars have been released (e.g., 'Prime-Jim®', 'Prime-Jan®', 'Prime-Ark® 45', 'Prime-Ark® Traveler', and 'Prime-Ark® Freedom') [2]. The PF blackberry 'Prime-Ark® Traveler' produces good shipping-quality fruit, and is recommended for the commercial fresh fruit market [7]. In the Southeastern United States (US) and coastal California, where the growing season is long and winter conditions are relatively mild and/or in high tunnels which allow for season extension, PF blackberries can be double-cropped with an early summer crop produced on over-wintered floricanes and on the current year's primocanes for a summer-to-fall crop. They can also be managed to fruit only on floricanes by mowing both the floricanes and primocanes, after fruiting is completed on the floricanes, and keeping the primocanes that emerge after harvest. Another option is to only fruit on the primocanes by pruning all canes after they have completed fruiting in late fall. In the following year, the fruit production occurs on the new primocanes that emerge in spring. In areas with severe winter conditions, there is less concern about winter damage if PF blackberries are grown to produce fruit only on the primocane. A primocane-only fruiting system has an added benefit of producing a crop into the fall when availability of fresh-market blackberries from Mexico and California becomes limited [8]. In the mid-Atlantic coast region, the cost of labor for summer cane tipping and hedging for PF blackberry to restrict plant height and to increase fruiting is considerably less than that for trellised floricane-fruiting blackberries, which requires selective pruning of primocanes to shorten the main and lateral canes in the summer and removal of floricanes in the winter [9]. Primocane management decisions are based on the environment in which the plants are growing, as well as the grower's marketing objectives, e.g., [1] producing fruit on over-wintered floricanes in early summer and on primocanes for late summer to fall, or [2] only on primocanes from late summer to fall [7].

Flower bud initiation in the apex of primocanes begins in late spring/early summer after primocanes have produced 20 nodes, or after two months of vegetative growth [10]. However, in upright primocanes of PF blackberry, axillary buds further down the cane generally do not break in the first growing season, but rather break the following season and produce flower shoots. Current primocane management practices for PF-only fruit cropping in PF blackberries include the removal of terminal growth on the primocanes in early summer and mowing of canes after harvest in the fall. The primocanes are either soft tipped, which is the removal of the most distal ~5 cm end of 0.7- to 1-m-tall primocanes, or hard tipped which is the removal of a longer portion of the cane once they are more mature. These practices are performed to increase branching from a few buds below the cut. These shoots then differentiate flower buds at their apices. Five to six weeks later, these buds reach anthesis, after which the berry development occurs. The fruit ripens in about seven weeks. This makes it possible to harvest fruit from mid-summer to first fall frost [4,11–13]. However, in the Southeastern US and areas with a wet, humid condition, the decapitation of actively-growing stout primocanes during the late spring to summer period can cause cane blight to develop, resulting in loss of productivity, or even cane death from infection through open pruning cuts [14]. In more northern regions with a shorter growing season, PF blackberry production has been limited, due to the short, late harvest season [15]. Also, a frost can damage fruit and end the harvest season before the crop reaches full maturity [16]. The development of alternative training methods for PF blackberries that can promote axillary bud break along the entire cane length, to increase fruit production on the primocanes, change the harvest season, and reduce cane blight infection, would greatly benefit growers.

In blackberries, the primocanes emerge in "flushes" from early spring to summer from adventitious buds on roots and latent buds on the crown [17]. The primocanes of FF blackberry from the early flush in April produced more lateral shoots than those that emerged in May and June [3,18]. Also, prior research in PF blackberries showed that delaying primocane growth by cutting the early flush growth and allowing fruiting on primocanes from later flush delayed fruiting and extended the harvest, compared to plants with uncut primocanes [19]. Changing branch orientation achieved by shoot or

cane bending has long been used to promote axillary bud break, and even flowering, in tree crops [20], grapevines [21], herbaceous plants [22], and FF blackberries [23,24]. In FF blackberry, a cane training system that included bending of ~70-cm tall primocanes, and forcing subsequent extension growth to occur horizontally, resulted in as many as ten long lateral canes on each primocane, compared to two or three laterals on upright, topped primocanes [3,18,25]. A preliminary study conducted with a thornless PF blackberry 'Prime-Ark Freedom' showed primocane bending and leaf removal stimulated lateral cane emergence from six or more axillary buds, which suggested that primocane bending and leaf removal promote development of laterals from axillary buds [24]. The observations made on thorny PF blackberry 'Prime-Ark 45' suggested that the primocane emergence occurred in two flushes: e.g., 10 or more vigorous primocanes from the first flush, and fewer, less vigorous, primocanes from the second flush. We hypothesized that, in thornless PF blackberries, cane bending and leaf removal treatments would enhance their cropping potential. The objectives of this research were to study primocane development from early spring to summer, and to examine the effects of bending and defoliation of early- and late-emerging primocanes on their reproductive development.

## 2. Materials and Methods

### 2.1. Experimental Location and Design

Tissue-cultured plug plants of 'Prime-Ark® Traveler' PF blackberry plants [7] were purchased from a commercial nursery in 2013 and established at the Appalachian Fruit Research Station in Kearneysville, WV (39.387° N, −77.886° E), located in the mid-Atlantic coast region of the Eastern United States. The blackberry plants were grown on raised beds covered with black landscape fabric. The nursery plants were transplanted through a 30 cm × 30 cm opening in the fabric, spaced 1.5-m apart, in rows spaced 3.4 m apart. The plants were drip irrigated as needed, and a granular 10-10-10 fertilizer was applied twice each season, at the rate of 200 g per plant each time, placed in the planting hole (270 kg/ha). The blackberry plants were grown using a modified T-trellis system with two 75-cm-wide cross-arms at 0.80 m and 1.15 m heights (Figure 1). Four wires were installed, evenly spaced, on the lower cross-arms. Two wires were installed on the upper cross-arms. Also, additional wires were installed 30-cm apart, to span the wires on the lower cross arms.

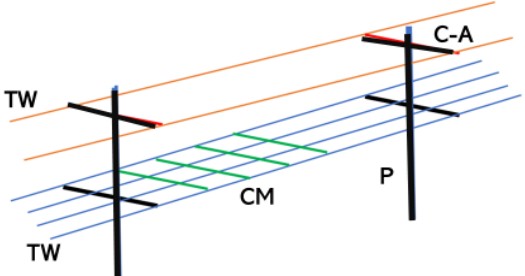

**Figure 1.** Schematic drawing (not to scale) of the modified T-trellis used in the study. The trellis system consisted of posts (P) and two cross arms (C-A) on each post. Bent primocanes were secured to one of four wires (TW, blue) on the lower cross-arms, and cross member (CM, green) wires were installed across these four wires. The wires on the upper cross-arms prevented the fruiting shoots from bending outward.

### 2.2. Experimental Description and Measurements of Generative and Reproductive Development

In the spring of 2016, four blocks of four plants of 'Prime-Ark®-Traveler' primocane-fruiting blackberry plants, that had been pruned during the dormant period, were selected to follow primocane cane development throughout the growing season from the time of cane emergence in early spring to harvesting of mature fruit later in the year. The number of primocanes emerged at the soil line was recorded at weekly intervals, and each emerged primocane was tagged with a color-coded label.

The first 10 primocanes to reach the wires on the lower cross arm were selected for further training. These primocanes were bent and secured to grow horizontally on one of the wires on the lower cross arm and the date of bending was recorded. These 10 primocanes were either soft tipped when a flower bud appeared at their terminal or the cane section along the training wire had extended another ≥0.7 m, whichever occurred first. All laterals developed from the axillary buds beyond the bend were retained, but those developing from the vertical portions of the canes (below the bend) were removed as they emerged. In addition, the total number of nodes above and below the bend, the number of secondary laterals to flower, the number of axillary buds to push above the bend, and the number of flowers on each primocane were recorded.

In 2017, primocane management and data collection described for 2016 were followed. In the second year of the study, four blocks (four plants per block) of 'Prime-Ark® Traveler' primocane-fruiting blackberry plants were randomly assigned one of two cane management treatments. Half of the plants were assigned to retain a maximum of 10 primocanes from the first flush, with the remaining canes or those that emerged subsequently pruned back thereafter, and the other half of the plants were assigned to retain five primocanes from the first flush (e.g., those that emerged in April) and five primocanes from the second flush (e.g., those that emerged in June), and the remainder of primocanes pruned back. On each plant, 10 primocanes were bent and secured to a training wire on the lower cross-arm to force extension growth to occur horizontally. For each primocane, the dates of their emergence and the date of bending were recorded. After cane bending, all leaves and secondary laterals developing from the vertical portions of the canes (below the bend (e.g., arc) in the primocane) were removed. Individual canes were soft tipped either when the terminal flower began to emerge, or the cane section along the training wire had extended ≥70 cm, whichever occurred first. All mature fruit were harvested, counted, and weighed to determine the yield per plant. The overall harvest duration for each plant was determined based on dates of first and last black fruit removed. Following the first hard frost at the end of October, total length of each trained canes, the total number of nodes above and below the bend, the number of secondary laterals to flower, and the number of axillary buds to push above the bend were recorded. The total number of flowers or berries per cluster, based on counts of calyx remnants, were also recorded. All data were tabulated to determine total flowers, total number of clusters, average cluster length, average flower number, and average cluster number per plant.

### 2.3. Statistical Analysis

The experiment in 2017 was performed using a split plot design with two main treatments (e.g., cane composition) and two subplot treatments (please change to "i.e." because the intent is to say "in effect" defoliation) with four blocks of four plants. All data were subjected to analysis of variance, with all percentage values transformed by an arcsin square root transformation prior to analysis. All data were separated either by *t*-test or DIFF option using SAS PROC MIXED at *p*-value of 0.05 [26].

## 3. Results

### 3.1. Primocane Development

The primocane emergence in PF "Traveler" blackberry occurred in two flushes in 2016 (Figure 2 and Table 1). The first flush (i.e., early) occurred in April, and 10 or more vigorous primocanes reached the trainable height, beginning in mid-May. Less than one primocane emerged from each plant in May. The second flush (here please change to i.e., late because the intent is to say "in effect") of primocane emergence occurred after 1 June, but only about five primocanes produced during the second flush reached the trainable height. Those primocanes that emerged in the latter half of April did not reach the trainable height until late June (Table 1). This suggests that there was a variation in their rate of growth and development. After these primocanes were bent and secured to a training wire on the lower cross arms, they continued to grow horizontally, and produced >1.4 m of growth from the

ground. The primocanes from the first flush that had reached the bendable height in May produced five lateral shoots from the axillary buds within the 0.7-long horizontally oriented section (Figure 3), and in early July, these shoots began to bloom. Also, these primocanes produced more and larger inflorescences, compared to those trained later (i.e., ~100 flowers per primocanes vs. < 70 flowers per primocane). Occasionally, there were laterals that emerged below the bend and unintentionally not removed by suckering. They grew vigorously, and eventually produced a large cluster of fruit at their terminals (Figure 4).

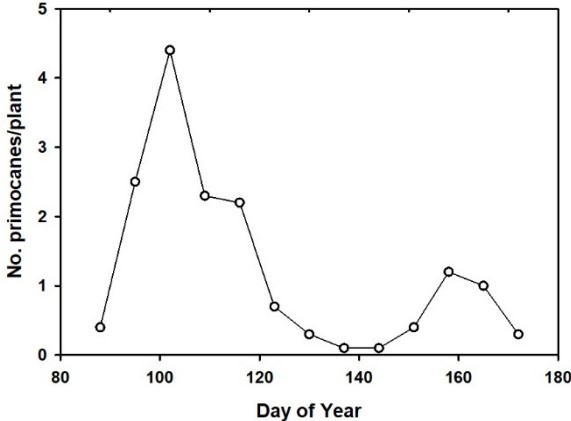

**Figure 2.** Primocane emergence from below the soil line from late March to end of June in "Traveler" blackberry in 2016. Newly emerged primocanes were counted and tagged at weekly intervals. Only the primocanes that reached the training wire are represented in the figure.

**Table 1.** Effects of primocane bending date on vegetative growth and reproductive development of 'Prime-Ark®Traveler', as illustrated by cane length to tip, number of inflorescences on each primocane, and numbers of flowers on each inflorescence (no. flowers/primocane) in 2016. Ten primocanes in this study were comprised of primocanes that emerged before 1 May on each plant.

| Cane Bending Date | Total Cane Length (cm) | Flower Shoots/Primocane (no.) | Flowers/Primocane (no.) |
|---|---|---|---|
| 17 May | 158 d [z] | 5.1 a | 107 a |
| 24 May | 154 d | 4.7 ab | 88 abcd |
| 31 May | 141 d | 3.5 c | 60 cde |
| 7 June | 142 d | 3.6 bc | 67 bcde |
| 14 June | 160 d | 4.1 abc | 90 abcd |
| 21 June | 151 d | 3.4 c | 44 de |
| 28 June | 150 d | 3.7 abc | 29 e |
| 5 July | 167 cd | 4.5 ab | 96 ab |
| 12 July | 188 ab | 4.1 abc | 67 bcde |
| 19 July | 178 bc | 2.5 c | 33 e |
| 26 July | 207 a | 3.0 c | 46 cde |
| *p* value | <0.0001 | 0.0521 | 0.0012 |

[z] Mean separation within columns by DIFF option of SAS [26] of PROC MIXED. Means within the same column with different letters are significantly different at the *p* = 0.05 level.

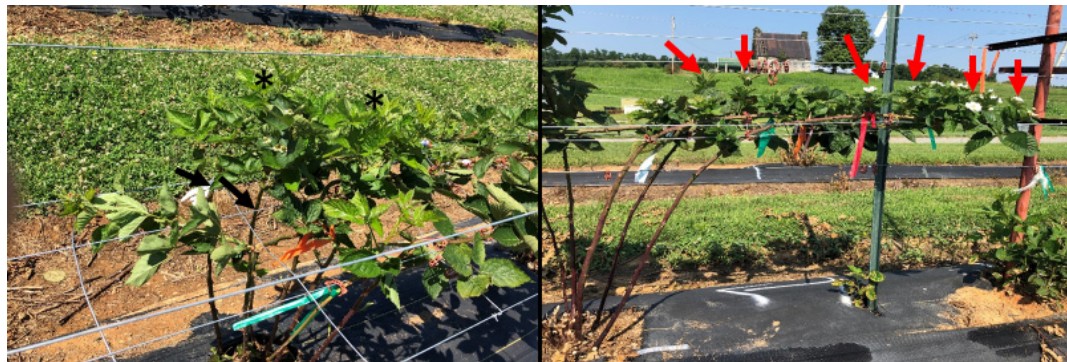

**Figure 3.** The development of bent primocanes on the trellis is illustrated with photographs taken in June (left) and July (right). The left photograph shows several primocanes that have been bent (arrows) and secured to training wires. Lateral shoots have emerged from the axillary buds at the bend and beyond (*). The right photograph shows three bent primocanes trained to the right after they have grown >0.7 m on the training wire. Note that many laterals have emerged and on some of these laterals flowers with white petals (red arrow) have developed at their terminals. All these laterals developed from the axillary buds located along the horizontal portion of bent primocanes. All the leaves and laterals below the bend were removed.

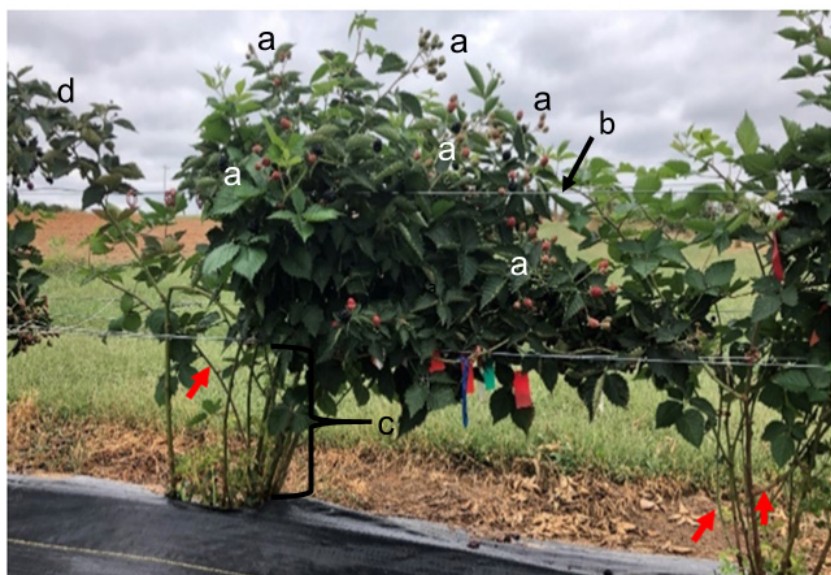

**Figure 4.** This image illustrates the maturing blackberry fruit on upright lateral shoots (a) that developed from axillary buds on bent primocanes. The primocanes were defoliated after cane bending. Note that fruiting shoots are confined within the two wires on the upper cross arms, with the wire (b) on one end of the cross arm shown), and the lack of leaves below the lower training wires (c). Also, note there are several laterals below the bend (arrows) that were unintentionally not suckered, because a complete suckering was not performed. Unexpectedly, these laterals grew >0.5 m and produced flower buds at their terminals (d). Photographed in August.

### 3.2. Effects of Primocane Bending Date and Defoliation

Primocane bending date and defoliation influenced lateral budbreak and length and on size of inflorescences (e.g., number of flowers on flowering shoots) (Table 2). Leaf removal treatment increased the number of lateral shoots, but they were shorter, and produced fewer flowers. Flower shoot numbers were more abundant on primocanes that had reached the trainable height by early July, compared to those that reached 1-m height after mid-July. The results also showed that the primocanes that were trained in May and in the first half of June had more flowers than those that were trained after mid-June. Thus, the plants with 10 primocanes that emerged in April had about 350 more flowers,

and were twice as productive than those plants that had the combination of early flush primocanes (e.g., emerged in April) and late flush primocanes (e.g., emerged in June).

**Table 2.** The effect of cane management techniques (e.g., selection of primocanes from early flush or those that emerged in April and from late flushes or those that emerged in June) and defoliation treatment (e.g., all leaves were removed or not removed from nodes on the horizontal portion of primocanes on lateral numbers per plant, lateral length (cm), flower numbers per shoot, number of fruits harvested, and plant yield for 'Prime-Ark® Traveler' primocane-fruiting blackberry. This study used 10 primocanes that emerged before 1 May (early flush) and a combination of 5 primocanes each from the early and late flushes.

| | Laterals/Plant (no.) | Lateral Length (cm) | Flowers/Shoot (no.) | Fruit/Plant (no.) | Yield/Plant (kg) |
|---|---|---|---|---|---|
| Primocane composition | | | | | |
| 10 early | 40 a | 48 a | 22 a [z] | 842 a | 4.7 a [z] |
| 5 early and 5 late flush | 39 a | 52 a | 17 b | 471 b | 2.6 b |
| Leaf treatment | | | | | |
| Defoliated | 49 a | 44 b | 15 b | 686 a | 3.9 a |
| Not defoliated | 30 b | 56 a | 24 a | 628 a | 3.4 a |
| *p* value | | | | | |
| Cane composition | 0.9541 | 0.0621 | 0.0009 | 0.0047 | 0.0091 |
| Leaf | 0.0055 | <0.0001 | <0.0001 | 0.5627 | 0.3974 |
| Cane composition × Leaf | 0.4341 | 0.0285 | 0.5499 | 0.1382 | 0.2273 |

[z] Mean separation within columns by *t*-test or DIFF option of SAS [26] of PROC MIXED. Means within the same column and in main plots and in subplots with different letters are significantly different at the *p* = 0.05 level.

Further analysis of flowering that occurred from July to mid-August on the primocanes that emerged in April showed that defoliation increased flower shoot numbers by 63% over non-defoliated primocanes, however, these shoots produced significantly fewer flowers. As a result, the defoliation treatment did not significantly increase plant yield (Table 2).

Primocane emergence occurred as early as the first of April, and primocanes continued to emerge into late June. Among the primocanes that emerged in early April (day 96), first bloom on non-defoliated plants was recorded on 29 June, while for flowering on primocanes that emerged in late April (day 124), anthesis was not recorded until day 227 (14 August) or 6 weeks later (Table 3). Defoliation treatment had little or no effect on bloom dates. Flowering in primocanes that emerged after late May was delayed by additional 3 to 4 weeks. By the third week in August, most of the primocanes that had not been defoliated had mature fruit (Figure 4), compared to only in 8% of primocanes that had been defoliated, thus defoliation delayed the peak harvest period. A similar yield response has been noted in FF blackberry [3,6] where primocanes that emerged early in the spring produced more fruit the following summer than later emerging primocanes.

**Table 3.** Effects of primocane emergence date in April (day 96, 103, 110, 117, and 124) and leaf defoliation treatment (all leaves removed, or defoliated, and no leaves removed, or not defoliated) after primocanes were tipped on bloom date. Primocanes that emerged in April were used in the study.

| | Primocane Emergence Date | | | | |
|---|---|---|---|---|---|
| | 96 | 103 | 110 | 117 | 124 |
| Leaf treatment | First bloom date | | | | |
| Not defoliated | 188 a [z] | 202 a | 210 ns | 218 ns | 227 ns |
| Defoliated | 196 b | 214 b | 210 | 215 | 215 |

[z] Mean separation within columns by *t*-test with SAS [26] PROC MIXED option at *p* = 0.05. Mean values that were not significantly different at the *p* = 0.05 level are indicated by ns.

## 4. Discussion

Primocanes that emerge in April and are not tipped can grow vertically ≥2.5 m height and produce one large inflorescence at their tip. On these upright primocanes, the axillary buds located below the distal one-third do not differentiate into reproductive buds in the summer [17]. Prior studies with PF blackberries have shown that pruning and tipping practices increase yield [8,12]. Pruning and tipping of primocanes at ~0.9 m height cause only two or three axillary buds below the cut to break and grow as lateral shoots [4]. The apical meristem in these lateral shoots has the potential to differentiate an inflorescence, but flower bud differentiation rarely occurs in buds located at nodes located at bottom two-third section of primocanes [17]. Unexpectedly, this study revealed that flower bud differentiation can occur in buds located along the entire length of primocanes, and produce fruit in the year of primocane emergence when actively growing primocanes are bent and whether or not they were defoliated (Figures 3 and 4). Tipping and pruning to terminate the vertical extension growth of the primocane can promote branches to develop from a few axillary buds below the cut [8,27]; whereas in this study, flower shoots developed all along the horizontal portions of bent primocanes. If laterals that emerged below the bend (e.g., vertical portion of primocane) were not removed by suckering, they, too, differentiated flower buds at their terminals. We also observed that the lateral shoots developing from nodes at, or near, the bend in the primocanes were the most productive.

More importantly, tipping and pruning or bending of primocanes allows fruit to develop closer to the ground so that harvesting can be managed more efficiently [4]. In the current study, primocanes were bent and trained to grow horizontally at ~0.7 to 0.8-m height. Up to ten lateral shoots that emerged from bent primocanes developed flowers at their terminals. Fruit production was confined to an area about 1.2- and 1.4-m high, and within the width of the upper cross-arm (Figures 3 and 4). Most fruiting laterals were upright, and mature fruit were not occluded by leaves. However, without support wires on the upper cross-arm, the inflorescence axis of upright fruiting shoots can kink as the fruit develop and gain weight. A kink in the inflorescence axis can lead to a disruption of photosynthate translocation to the developing fruit, which could prevent the fruit from properly maturing (Takeda, personal observation). In contrast, when upright primocanes are tipped, the laterals that emerge from them radiate from the canes inwardly and outwardly, relative to the row middle. Harvesting of fruit from laterals that had developed inward becomes less efficient as the leaves on nearby fruiting shoots can obscure the fruit from the picker. Also, numerous primocanes create a dense canopy, especially if the leaves at the base of the plant are not defoliated. The dense canopy could provide a good resting area for insect pests, such as the spotted wing drosophila (*Drosophila suzukii* Matsumura) (SWD), during the hotter parts of the day [28]. The defoliation of leaves below the bend could make the bottom half of the plant less hospitable for SWD.

The results of this study indicate that our cane manipulation technique (e.g., bending and defoliation) altered the development of shoots from the axillary buds along the entire length of bent primocanes (e.g., horizontally oriented, distal section, bent section, and below the bent section). The increase in flower shoot numbers on early emerging primocanes can potentially contribute to improving plant yield from the primocanes. However, with increased numbers of laterals emerging in summer and fall, fewer unbroken buds remained for fruiting the following summer from over-wintered floricanes; however, in California, the canes of PP blackberries are no longer over-wintered to obtain fruit on their floricanes, because early fruiting FF cultivars with high fruit quality are available to growers (E. Thompson, Pacific Berry Breeding, 2020 Personal communication). If a satisfactory yield can be obtained from the primocane-only production system, growers can simply mow down the fruited primocanes after the season, without need for over-wintering canes [29], for fruit production on the floricane the following season. Both FF and PF blackberries are adaptable to various production systems, as different trellis and cane training techniques are used for their production [3,4,18,20]. The tipping of primocanes in early summer removes apical dominance and encourages branch formation from axillary buds. Axillary bud development can also be manipulated with chemical treatments. In Mexico, FF "Tupy" blackberry is grown using plant growth regulators and a chemical-based system in which

potassium sulfate ($K_2SO_4$) or similar salts are sprayed about five times at weekly intervals in the summer to force leaf desiccation, abscission, and axillary bud break [2,30], which extends the fruit production period. Under the tropical and subtropical growing conditions of Brazil, hydrogen cyanamide is used to overcome insufficient cold temperatures and promote budbreak, flowering, and fruit production [31]. In eastern thornless blackberries, yield is increased by having plants produce more cane length and bud numbers by bending primocanes that promote axillary budbreak [3,6,32]. In this study, alternative cane management practices were evaluated for a newly released PF blackberry. Primocane bending and defoliation resulted in a 60% increase in the number of flower shoots per primocane, compared to the non-defoliated primocanes. Although flower shoots were more abundant in defoliated primocanes, fruit numbers were less, or remained unchanged. Additional studies are needed to investigate these new primocane management techniques (e.g., primocane bending at lower height) to increase lateral branch cane growth and flower numbers.

Upright primocanes of PF blackberries can grow >2.5-m-tall [15] and produce fruit usually only at their terminal. However, by bending the primocanes about ~20-cm from the tip when the primocanes are 1-m-tall, their subsequent extension growth occurs horizontally by periodically securing their distal ends to the training wires. When the primocanes are horizontally oriented, there is no strong dominance by one bud over other axillary buds, resulting in more lateral shoots emergence along the entire length of bent primocanes. An alternative method of primocane manipulation would be to allow the primocanes to reach a height of 2.0 m or more, and then bend them horizontally at a height of about 0.7 m, which would reduce the time needed for cane training. Either method will require bending of primocanes and each cane would ~1.3 m section that is oriented horizontally. Each one of those axillary buds developed at 15 to 20 nodes at the bend and along the horizontally oriented portion of primocane has the potential to develop a flower shoot in the current year, rather than in spring of the following year. The bent primocanes could also be defoliated mechanically or chemically, following cane bending, to improve bud break. However, additional treatment protocol is needed to increase flower numbers on them.

A production method that avoids the need for cane tipping would be desirable in some areas. In the Southeastern US, cane blight is a serious fungal disease [14]. The spores of the causal agent *Leptosphaeria coniothyrium* enter primocanes through recently injured and pruned cuts from the spring to the fall, and even cause the entire cane to decay. The current recommendation for controlling this disease is to apply fungicides after each pruning to provide a protective barrier on the wound site. Thus, production practices, such as the cane bending method that promotes flowering without the need to prune actively growing primocanes, could contribute to reducing infection sites.

According to Gaskell and Daugovish [27], the most favorable market window for fresh blackberries is from mid-June to early December. This period coincides with reduction of domestic production from FF blackberries and increased importation of blackberries from Mexico [2]. Using current production practices in the coastal regions of California, the harvest peaks for PF blackberries are from late August to early September. Growers are interested in pruning and cane training practices that may permit greater harvest volumes during the September to December period. Pruning and training practices described in this study increased flower numbers and delayed flowering, which may offer a means to obtain greater yields later in the season. Additional studies are needed to determine what other cultural practices (e.g., cane manipulation and/or plant growth regulator applications) will promote bud break and lateral shoot development in PF blackberries and alter the primocane development for greater fruit production from the late summer to the fall period.

## 5. Conclusions

This study investigated the effects of primocane orientation and defoliation treatments on the reproductive potential of PF blackberry. To our knowledge, this study is the first to describe the relationships between primocane emergence date, vigor, orientation, and leaf removal on plant productivity in PF blackberry. In this study, the primocane management practice focused on cane

bending and defoliation, which led to increased flower shoot numbers. The findings suggest that these practices singularly, or in combination, have the potential to increase fruit production in PF blackberries. Another relevant outcome of this study was the realization that more axillary buds along the entire length of primocanes have the capacity to develop a flower shoot in the current growing season. This study provided new clues for future investigations on cane manipulation techniques for improving the productivity of PF blackberry. These efforts will further our knowledge of how cane manipulations can alter the growth and development of axillary buds. Knowing how cane orientation affects reproductive development can enhance our understanding about the regulation of yield potential in PF blackberry, and lead to refinements in the trellis design to optimize primocane growth and reproductive development.

**Author Contributions:** Conceived the experiments: F.T.; designed the experiments: F.T. and A.R.; performed the experiments: F.T. and A.R.; analyzed the data: A.R. and K.D.; contributed reagents/materials/analysis tools: F.T.; wrote the paper: F.T. and A.R. All authors have read and agreed to the published version of the manuscript.

**Funding:** This research was funded by USDA-ARS National Program 305 Crop Protection and Production and by USDA ARS in-house Project No. 8080-21000-025-00D "Small fruit production research" and was partially supported by a grant from the North American Bramble Growers Research Foundation, Pittsboro, NC 27312, USA (www.raspberryblackberry.com).

**Acknowledgments:** Special appreciation is extended to Breyn Evans, Wade Snyder, and Tony Rugh for their assistance.

**Conflicts of Interest:** The authors have no conflict of interest to report.

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
