# Peer review of "Effects of Cane Emergence Time, Bending, and Defoliation on Flowering and Yield in Primocane-Fruiting Blackberry"

_agronomy, doi:10.3390/agronomy10111737_

Round 1

Reviewer 1 Report

In general, this is an interesting paper with a good justification for the work. However, the defoliation treatment referred to is not clearly described, thus it is difficult to understand exactly what the authors did and what they based their conclusions on. Additionally, although the English is good, there are numerous typos and partial sentences throughout the manuscript that make understanding difficult. Specific comments:

Lines 140-141 - Is this the defoliation treatment the authors refer to? The removal of buds below the bend? This is unclear, but central to the work, so must be clarified.

Line 164 - What were the main plots and subplots?

Line 175 - Unclear how Table 1 shows that PC emerging in latter half of April didn't reach trainable height until July

Line 175 - Example of partial sentence that obscures the meaning of what the authors are trying to say. This occurs throughout the m.s.

Figure 2 - Add month to x-axis DOY legend, since the authors present the data here in terms of month in the narrative.

Line 188-189 - So, is this the intact control for the defoliation treatment (i.e. "Occasionally, there were laterals inadvertently left unpruned.")? This is the only reference I could find to an intact treatment, yet this gives the decided impression that the "control" treatment was accidental. What am I missing here?

Line 215 - Doesn't Table 1 show these data? Or is Table 1 only for year 1 and this paragraph refers to Year 2? This isn't clear.

Line 219 - What is "...early and flush primocanes"?

Table 2 - Again, please clarify in M&M and throughout what the "Intact" treatment was and what the "Defoliated" treatment was. How many of each were sampled (what was "n")?

Lines 274-275 - This contradicts my understanding of what was meant by defoliation, which I though alluded to the removal of lateral buds below the bend. But here, the authors state that bending and defoliation increase lateral bud break along the cane, including below the bend. So now I don't know what defoliation is referring to.

Lines 285-290 - Incomplete sentences (another example) 

Author Response

Please see attached document that has response to both reviewers.  

Reviewer 2 Report

Methodology. Add the number of plants per treatment in each subplot.

Author Response

Please see attached document with our response to all questions and comments from the review team.
